# Clinical Link between the BARD Score at Diagnosis and Mortality during Follow-Up in Patients with Antineutrophil Cytoplasmic Antibody-Associated Vasculitis

**DOI:** 10.3390/jcm12175679

**Published:** 2023-08-31

**Authors:** Tae-Geom Lee, Pil-Gyu Park, Yong-Beom Park, Ji-Hye Huh, Sang-Won Lee

**Affiliations:** 1Department of Medicine, Yonsei University College of Medicine, Seoul 03722, Republic of Korea; taegeom.lee19@med.yuhs.ac; 2Division of Rheumatology, Department of Internal Medicine, National Health Insurance Service Ilsan Hospital, Goyang 10444, Republic of Korea; pilgyupark@nhimc.or.kr; 3Division of Rheumatology, Department of Internal Medicine, Yonsei University College of Medicine, Seoul 03722, Republic of Korea; yongbpark@yuhs.ac; 4Institute for Immunology and Immunological Diseases, Yonsei University College of Medicine, Seoul 03722, Republic of Korea; 5Division of Endocrinology and Metabolism, Department of Internal Medicine, Hallym University Sacred Heart Hospital, Anyang 14068, Republic of Korea

**Keywords:** BARD score, antineutrophil cytoplasmic antibody, vasculitis, mortality

## Abstract

This study investigated whether the BARD score at diagnosis could predict all-cause mortality in patients with antineutrophil cytoplasmic antibody-associated vasculitis (AAV). This study included 236 immunosuppressive drug-naïve patients without chronic liver diseases such as viral hepatitis, non-alcoholic fatty liver disease (NAFLD), and advanced liver diseases and their clinical data at diagnosis, such as age, sex, and the Birmingham Vasculitis Activity Score (BVAS). The BARD score was calculated by the sum of aspartate transaminase (AST)/alanine transaminase (ALT) ratio ≥ 0.8 (+2 points), body mass index (BMI) ≥ 28 kg/m^2^ (+1 point), and the presence of type 2 diabetes mellitus (T2DM) (+1 point). All-cause mortality was investigated as a poor outcome of AAV. The median age of AAV patients was 60.0 years, and 34.7% were men. Among AAV patients, 7, 50, and 187 scored 1, 1, and 2 points owing to having a BMI ≥ 28 kg/m^2^, T2DM, and an AST/ALT ratio ≥ 0.8, respectively. Patients with a BARD score ≥ 2 and those with a BARD score ≥ 3 exhibited significantly lower cumulative patient survival rates than those without (*p* = 0.038 and *p* = 0.003, respectively). In the multivariable Cox analysis, among the two cut-offs of the BARD scores, only a BARD score ≥ 3 (HR 2.866), along with age (HR 1.061), male sex (HR 2.327), and BVAS (HR 1.100), was independently associated with all-cause mortality during follow-up. In conclusion, this study was the first to demonstrate that the BARD score ≥ 3 at AAV diagnosis could predict all-cause mortality during follow-up in AAV patients.

## 1. Introduction

The BARD score was first proposed by Harrison et al. for identifying non-alcoholic fatty liver disease (NAFLD), more conveniently compared with previous indices for NAFLD [1]. The BARD score is composed of three parameters, namely, the aspartate transaminase (AST)/alanine transaminase (ALT) ratio, body mass index (BMI), and type 2 diabetes mellitus (T2DM) [1]. A summation of the BARD score ranges from 0 to 4, and a total score of ≥2 was associated with advanced liver fibrosis [1,2,3]. In addition to the advantage of simplifying the score by setting the interval, the BARD score has the advantage that it showed high sensitivity and specificity similar to other NAFLD-reflecting indices and a high negative predictive value compared with others despite the relatively low accuracy [4,5].

Antineutrophil cytoplasmic antibody (ANCA)-associated vasculitis (AAV) is one of the autoimmune systemic vasculitides and primarily affects small vessels, such as capillaries and adjacent arterioles and venules [6]. AAV is categorised into three subtypes, namely, microscopic polyangiitis (MPA), granulomatosis with polyangiitis (GPA), and eosinophilic granulomatosis with polyangiitis (EGPA), according to clinical, laboratory, radiological, and histological features [6,7]. Several patients with AAV experience unwanted, poor outcomes of AAV during follow-up; thus, it is pivotal to determine indices at AAV diagnosis for predicting poor AAV outcomes during follow-up [8].

A previous study attempted to demonstrate the predictive potential of the baseline values of NAFLD and liver fibrosis-related indices for all-cause mortality during follow-up in AAV patients [9]. Given that the parameters composing the formula of the BARD score, including BMI, liver enzyme levels, and T2DM, are closely associated with mortality in the general population [10,11,12], it could be reasonably speculated that the BARD could predict all-cause mortality in patients with AAV. Nevertheless, no study has evaluated the predictive potential of the baseline BARD score for all-cause mortality until now. Thus, this study aimed to investigate whether the baseline value of the BARD score at AAV diagnosis could predict all-cause mortality in patients with AAV.

## 2. Materials and Methods

### 2.1. Study Subjects

This study selected 236 immunosuppressive drug-naïve AAV patients without chronic liver diseases from an observational single-centre cohort of AAV patients, i.e., the Severance Hospital ANCA-associated Vasculitides (SHAVE) cohort, according to the following inclusion criteria: (i) those who were first classified as having MPA, GPA, and EGPA at Yonsei University College of Medicine, Severance Hospital, by specialised rheumatologists; (ii) those who fulfilled the 1990 American College of Rheumatology criteria for EGPA, the 2007 European Medicine Agency algorithm for AAV, and the 2012 revised Chapel Hill Consensus Conference nomenclature of vasculitides [6,7,13]; (iii) those who had sufficient data at AAV diagnosis, including AAV subtypes, ANCA type, Birmingham Vasculitis Activity Score (BVAS), and five-factor score (FFS) [14,15]; (iv) those who essentially had data at AAV diagnosis regarding the parameters of an equation for calculating the BARD scores such as BMI, AST/ALT ratio, and T2DM [1]; (v) those whose data regarding all-cause mortality were clearly documented in medical records at the recent visit; (vi) those who did not have chronic liver diseases such as viral hepatitis, NAFLD and advanced liver diseases; (vii) those who had no serious medical conditions such as malignancies, infection diseases requiring close monitoring, and autoimmune systemic diseases mimicking AAV; (viii) those who had been followed for ≥3 months after AAV diagnosis; (ix) those who did not receive glucocorticoids (≥20 mg/day equivalent to prednisolone), or immunosuppressive drugs within one month before AAV diagnosis; and (x) those who signed an informed consent form at enrolment into the SHAVE cohort.

### 2.2. Clinical Data

Regarding variables at AAV diagnosis, age, male and female sexes, BMI, and ex-smokers were collected as demographic data, and AAV subtype, ANCA type, BVAS, and FFS were collected as AAV-specific data. Laboratory results included acute phase reactants and the AST/ALT ratio. T2DM was also reviewed for a formula for the BARD score. Regarding variables during follow-up, the number of patients who died and received glucocorticoids and immunosuppressive drugs was recorded.

### 2.3. ANCA Measurement

Perinuclear (P)-ANCA and cytoplasmic (C)-ANCA were detected using an indirect immunofluorescence assay, whereas myeloperoxidase (MPO)-ANCA and proteinase 3 (PR3)-ANCA were determined using an immunoassay. An immunoassay was primarily performed as a screening method; however, P-ANCA and C-ANCA were considered to have MPO-ANCA or PR3-ANCA when AAV was strongly suspected based on clinical and laboratory features [16].

### 2.4. Formula of the BARD Score

The formula for the BARD score includes three parameters, and differently weighted points are assigned to the higher range or presence of each parameter. Cases with an AST/ALT ratio ≥ 0.8, a BMI ≥ 28 kg/m^2^, and the presence of T2DM were assigned 2, 1, and 1 points, respectively [1].

### 2.5. All-Cause Mortality

In the present study, we defined all-cause mortality as death from any cause. The follow-up duration based on all-cause mortality was defined as the period between AAV diagnosis and death in deceased patients. Conversely, in surviving patients, the follow-up duration based on all-cause mortality was defined as that between the AAV diagnosis and the last visit [10,11].

### 2.6. Statistical Analyses

All statistical analyses were performed using IBM SPSS Statistics for Windows, version 26 (IBM Corp., Armonk, NY, USA). Continuous variables are expressed as medians with 25–75 percentiles, whereas categorical variables are expressed as numbers (percentages). A comparison of the cumulative survival rates between the two groups was analysed by the Kaplan–Meier survival analysis with the log-rank test. The multivariable Cox hazard model using variables with statistical significance (*p* < 0.1) in the univariable Cox hazard model was conducted to appropriately obtain the hazard ratios (HRs) during the considerable follow-up duration. *p*-values < 0.05 were considered statistically significant.

## 3. Results

### 3.1. Characteristics of AAV Patients

Regarding variables at AAV diagnosis, the median age of AAV patients was 60.0 years. Of the 236 patients, 34.7% were males and 65.3% were females. Among the 236 patients, 124, 62, and 50 had MPA, GPA, and EGPA, respectively. The median BVAS, FFS, erythrocyte sedimentation rate (ESR), and C-reactive protein (CRP) were 12.0, 1.0, 57.0 mm/h, and 13.5 mg/L, respectively. Among AAV patients, 7, 50, and 187 scored 1, 1, and 2 points owing to having a BMI ≥ 28 kg/m^2^, T2DM, and an AST/ALT ratio ≥ 0.8, respectively. The median BARD score was 2.0, while 189 (80.1%) and 40 (16.9%) patients had BARD scores ≥ 2 and ≥3, respectively. Regarding variables during follow-up, 28 patients died during an average follow-up duration based on all-cause mortality of 33.8 months. Glucocorticoids were administered to 221 (93.6%), and cyclophosphamide was the most frequently administered immunosuppressive drug (55.1%), followed by azathioprine (53.4%) (Table 1).

### 3.2. Comparison of the Cumulative Survival Rates

When patients were divided into two groups according to a BARD score of 2, patients with a BARD score ≥ 2 exhibited a significantly lower cumulative patient survival rate than those with a BARD score < 2 (*p* = 0.038). Additionally, patients with a BARD score ≥ 3 show a significantly reduced cumulative patient survival rate compared to those with a BARD score < 3 (*p* = 0.003) (Figure 1).

### 3.3. Cox Hazards Model Analyses

Given the clinical significance of the variables at diagnosis and the small number of patients in this study, the variables with *p* < 0.1 in the univariable Cox analysis were included in the multivariable Cox analysis. In the univariable Cox analysis, both a BARD score ≥ 2 (HR 6.307, *p* = 0.071) and a BARD score ≥ 3 (HR 3.267, *p* = 0.005) were significantly associated with all-cause mortality during follow-up, along with age (HR 1.086, *p* < 0.001), male and female sexes (HR 2.094 and 0.478, respectively, *p* = 0.053), BVAS (HR 1.117, *p* < 0.001), FFS (HR 2.208, *p* < 0.001), ESR (HR 1.009, *p* = 0.058), and CRP (HR 1.008, *p* = 0.004). Although both male and female sexes exhibited a tendency to be associated with all-cause mortality, because male sex shows a proportional association compared to an inverse association with female sex, male sex was selected and included in the multivariable Cox analysis. In the multivariable Cox analysis with a BARD score ≥ 2, age (HR 1.060, 95% confidence interval (CI) 1.015–1.108), male sex (HR 2.698, 95% CI 1.181–6.160), and BVAS (HR 1.087, 95% CI 1.025–1.154) were independently associated with all-cause mortality during follow-up but not a BARD score ≥ 2. Whereas, in the multivariable Cox analysis with a BARD score ≥ 3, a BARD score ≥ 3 (HR 2.866, 95% CI 1.175–6.991), together with age (HR 1.061, 95% CI 1.015–1.109), male sex (HR 2.327, 95% CI 1.035–5.233), and BVAS (HR 1.100, 95% CI 1.035–1.168), was independently associated with all-cause mortality during follow-up (Table 2).

## 4. Discussion

This study revealed that the BARD score at diagnosis, which is a clinical index for identifying NAFLD and reflecting advanced liver fibrosis, could predict all-cause mortality during follow-up in patients with AAV. By the way, this study included AAV patients who had not been diagnosed with significant chronic liver diseases. Nevertheless, a considerable number of patients had a BARD score ≥ 2 (80.1%) and a BARD score ≥ 3 (16.9%), which represent substantial NAFLD or advanced liver fibrosis. Therefore, the important message of this study is that the BARD score could be a useful indicator that can predict all-cause mortality in AAV patients by firstly reflecting the level of the inflammatory burden of AAV and, secondly, reflecting the extent of subclinical NAFLD or advanced liver fibrosis caused by inflammation-induced insulin resistance.

We analysed which parameter had the greatest effect on all-cause mortality in AAV patients among the three variables constituting the formula of the BARD score: AST/ALT ratio ≥ 0.8, BMI ≥ 28 kg/m^2^, and the presence of T2DM, using the Kaplan–Meier survival analysis with the log-rank test. We found that patients with an AST/ALT ratio ≥ 0.8 exhibited a significantly lower cumulative patient survival rate than those with an AST/ALT ratio < 0.8 (*p* = 0.033). Additionally, patients with T2DM show a significantly reduced cumulative patient survival rate compared to those without T2DM (*p* = 0.041). However, BMI ≥ 28 kg/m^2^ was not likely to contribute to the significant differences in the cumulative patients’ survival rates (*p* = 0.778) (Appendix A).

Regarding the AST/ALT ratio, it can be assumed that the two mechanisms may be involved in the possibility to predict all-cause mortality in AAV patients. The first assumption is that the AST/ALT ratio could predict all-cause mortality by reflecting tissue damage not confined to the liver, or, in other words, reflecting the level of the inflammatory burden of AAV. The AST/ALT ratio has been considered a significant blood circulation indicator for predicting the prognosis of a wide variety of chronic systemic inflammatory diseases [17,18,19]. While ALT is thought to have a stronger hepatic specificity or abundant expression in liver tissues, AST has been discovered to be significantly expressed in the brain, muscle, and kidney tissues [20]. Therefore, the AST/ALT ratio has the potential to be a desirable indicator reflecting systemic pathological conditions beyond the liver. The second assumption is that the AST/ALT ratio could predict all-cause mortality by reflecting the extent of subclinical NAFLD or advanced liver fibrosis caused by inflammation-induced insulin resistance. Previous studies reporting that metabolic syndrome and NAFLD were associated with all-cause mortality in AAV patients may support this assumption [9,21].

Since T2DM is a well-known risk factor for mortality in the general population, the result that the presence of T2DM was associated with all-cause mortality in AAV patients was consistent with what was expected [12]. However, although the association of BMI with mortality shows a U-shape pattern, the result that BMI ≥ 28 kg/m^2^ was not associated with all-cause mortality was unexpected because BMI ≥ 28 kg/m^2^ is definitely significantly correlated with the rate of mortality [10]. We attribute this discrepancy to the small number of patients (only seven patients) having a BMI ≥ 28 kg/m^2^. It is believed that if the number of patients belonging to this range increased, a significant association would have been evident. Therefore, based on this theoretical basis, although a parameter of BMI ≥ 28 kg/m^2^ did not affect all-cause mortality, this study demonstrates that the BARD score, including the parameters of AST/ALT ratio ≥ 0.8 and the presence of T2DM, could predict all-cause mortality in AAV patients.

A BARD score ≥ 2 was reported to be associated with advanced fibrosis with an odds ratio of 17 [1]. However, the Cox hazards model analyses in this study proved that a BARD score ≥ 3 rather than a BARD score ≥ 2 was significantly and independently associated with all-cause mortality in AAV patients. This result may support the assumption that the BARD score reflects the inflammatory burden of chronic systemic diseases in addition to the extent of NAFLD or advanced liver fibrosis. In addition to the BARD score ≥ 3, age and male sex, well-known mortality risk factors in the general population, had an independent association with all-cause mortality in AAV patients, which may also support and enhance the reliability of the results of this study [22].

The absence of imaging tests for NAFLD and advanced liver fibrosis by non-invasive liver stiffness measurement (LSM) using transient elastography (FibroScan^®^, EchoSens, Paris, France) was a critical limitation of this study [23]. If the correlation between the results of imaging tests and the BARD score had been assessed, it could have been differentiated whether the association of the BARD score with all-cause mortality in AAV patients was due to subclinical deterioration in liver function or to an increase in insulin resistance caused by an inflammatory burden of AAV [9]. However, in real clinical practice, the coincidental diagnosis of AAV or misdiagnosis of AAV owing to ANCA false positivity in patients with primary autoimmune hepatitis, primary biliary cirrhosis, and primary sclerosing cholangitis have been reported occasionally [24]; however, the direct liver involvement of AAV has rarely been reported to date. Moreover, among the systemic items of the BVAS for assessing AAV activity, there is no item regarding hepatic manifestation [14]. Therefore, given that this study excluded AAV patients who had autoimmune systemic diseases mimicking AAV and who received glucocorticoids (≥20 mg/day equivalent to prednisolone) or immunosuppressive drugs within one month before AAV diagnosis, it may be concluded that most AAV patients in this study might not have clinically significant NAFLD or advanced liver diseases, and furthermore, the necessity of transient elastography performance could be offset relatively.

At the entry of the present study, the premise was that the BARD score at AAV diagnosis may not reflect the extent of NAFLD progression in patients with AAV but the simultaneous inflammatory burden, which subsequently may be associated with all-cause mortality during follow-up. In order to demonstrate this assumption and find out by which variables among AAV-specific indices and the general acute-phase reactants the BARD score can predict all-cause mortality, we investigated the correlation of the BARD score with BVAS, FFS, ESR, and CRP using the Pearson correlation analysis. We found that only BVAS of the four variables was significantly correlated with the BARD score at AAV diagnosis (r = 0.149, *p* = 0.022). In addition, given the possibility that it might not be appropriate to compare correlation coefficients among the BARD score and other circulating inflammatory biomarkers because the BARD score is composed of integers rather than continuous numbers, we compared the median levels of BVAS, FFS, ESR, CRP, and serum albumin between patients with a BARD score > 3 and those with a BARD score < 3. Patients with a BARD score ≥ 3 tended to exhibit higher median levels of BVAS than those with a BARD score < 3 (14.5 vs. 12.0, *p* = 0.054). However, there were no significant differences in FFS (*p* = 0.139), ESR (*p* = 0.185), or CRP (*p* = 0.124) between patients with a BARD score ≥ 3 and those with a BARD score < 3. Although the median levels of BVAS between the two groups divided by the BARD score ≥ 3 did not show clear statistical significance, they were close to significance with a *p* value of 0.054, and in the correlation analysis, the BARD score was significantly correlated with BVAS. Therefore, we conclude that, not totally but partially, the predictive potential of the BARD score at AAV diagnosis for all-cause mortality might be possible via reflecting the inflammatory burden represented by BVAS.

This study has several limitations. The number of patients was not sufficient to enable the generalisation of the results of this study to most patients with AAV. The retrospective study design was another hurdle to generalising the clinical significance of the BARD score in AAV patients. As mentioned above, another critical limitation was the absence of imaging tests for NAFLD, or advanced liver diseases. Additionally, further investigation of the association of the BARD score with death according to the causes of death may clarify the clinical usefulness of the BARD score and enhance its significance in real clinical settings. However, we could not perform it due to the clinical limitation that it is often difficult to clearly distinguish the causes of death in deceased patients with AAV among the worsening of the disease, side effects of immunosuppressive treatment, and serious infections. This was why we defined all-cause mortality as death of any cause in the present study. Nevertheless, this study has merit in that it was the first to demonstrate that the BARD score at AAV diagnosis could predict all-cause mortality during follow-up in AAV patients. Future prospective and multi-centre studies with more AAV patients can validate the results of this study and provide a more reliable clinical significance of the BARD score in AAV patients.

## 5. Conclusions

The present study was the first to demonstrate that a BARD score ≥ 3 at AAV diagnosis could predict all-cause mortality during follow-up in AAV patients. We suggest that the physicians use the BARD score at AAV diagnosis as a complementary predictor of all-cause mortality and pay more attention to AAV patients with an initial BARD score ≥ 3 during follow-up.

## Figures and Tables

**Figure 1 jcm-12-05679-f001:**
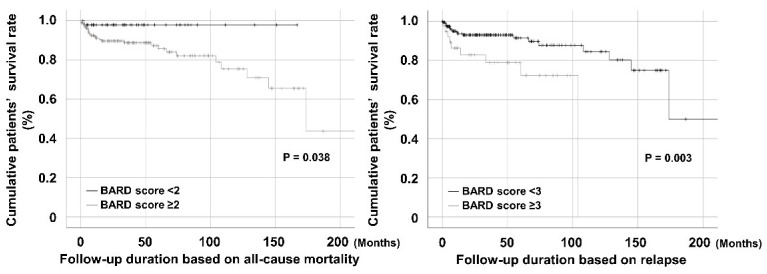
Comparison of the cumulative survival rates. Patients with a BARD score ≥ 2 and those with a BARD score ≥ 3 exhibited significantly lower cumulative patient survival rates than those without, respectively.

**Table 1 jcm-12-05679-t001:** Characteristics of AAV patients at diagnosis and during follow-up (N = 236).

Variables	Values
** *At AAV diagnosis* **	
**Demographic data**	
Age (years)	60.0 (48.3–69.0)
Male sex (N, (%))	82 (34.7)
Female sex (N, (%))	154 (65.3%)
BMI (kg/m^2^)	22.4 (20.1–24.2)
Ex-smoker (N, (%))	13 (5.5)
**AAV subtype (N, (%))**	
MPA	124 (52.5)
GPA	62 (26.3)
EGPA	50 (21.2)
**ANCA type and positivity (N, (%))**	
MPO-ANCA (or P-ANCA) positivity	157 (66.5)
PR3-ANCA (or C-ANCA) positivity	41 (17.4)
**AAV-specific indices**	
BVAS	12.0 (7.0–18.0)
FFS	1.0 (0–2.0)
**Acute phase reactants**	
ESR (mm/h)	57.0 (21.3–96.0)
CRP (mg/L)	13.5 (1.6–67.6)
**Liver-related variables**	
AST (IU/L)	18.0 (15.0–24.0)
ALT (IU/L)	16.0 (11.0–25.0)
AST/ALT ratio	1.2 (0.9–1.6)
**BARD score-related variables**	
BMI ≥ 28 kg/m^2^ (N, (%))	7 (3.0)
T2DM (N, (%))	50 (21.2)
AST/ALT ratio ≥ 0.8 (N, (%))	187 (79.2)
BARD score	2.0 (2.0–2.0)
BARD score ≥ 2 (N, (%))	189 (80.1)
BARD score ≥ 3 (N, (%))	40 (16.9)
** *During the follow-up duration* **	
**Typical poor outcomes of AAV**	
All-cause mortality (N, (%)	28 (11.9)
Follow-up duration based on all-cause mortality (months)	33.8 (9.9–68.4)
**Medications (N, (%))**	
Glucocorticoids	221 (93.6)
Cyclophosphamide	130 (55.1)
Rituximab	38 (16.1)
Mycophenolate mofetil	33 (14.0)
Azathioprine	126 (53.4)
Tacrolimus	20 (8.5)
Methotrexate	24 (10.2)

Values are expressed as a median (25–75 percentiles) or N (%). AAV: ANCA-associated vasculitis; ANCA: antineutrophil cytoplasmic antibody; BMI: body mass index; MPA: microscopic polyangiitis; GPA: granulomatosis with polyangiitis; EGPA: eosinophilic granulomatosis with polyangiitis; MPO: myeloperoxidase; P: perinuclear; PR3: proteinase 3; C: cytoplasmic; BVAS: Birmingham vasculitis activity score; FFS: five-factor score; ESR: erythrocyte sedimentation rate; CRP: C-reactive protein; AST: aspartate aminotransferase; ALT: alanine aminotransferase; T2DM: type 2 diabetes mellitus; BARD: BMI, AST/ALT ratio, and DM.

**Table 2 jcm-12-05679-t002:** Cox hazards model analyses of variables at diagnosis for all-cause mortality during follow-up in AAV patients.

**Variables**	**Univariable**			**Multivariable** **(BARD Score ≥ 2)**			**Multivariable** **(BARD Score ≥ 3)**		
	**HR**	**95% CI**	***p* Value**	**HR**	**95% CI**	***p* Value**	**HR**	**95% CI**	***p* Value**
Age (years)	1.086	1.044–1.130	<0.001	1.060	1.015–1.108	0.009	1.061	1.015–1.109	0.009
Male sex	2.094	0.989–4.431	0.053	2.698	1.181–6.160	0.019	2.327	1.035–5.233	0.041
Female sex	0.478	0.226–1.011	0.053						
Ex-smoker	2.045	0.615–6.797	0.243						
MPO-ANCA (or P-ANCA) positivity	1.776	0.750–4.207	0.191						
PR3-ANCA (or C-ANCA) positivity	0.357	0.085–1.511	0.162						
BVAS	1.117	1.064–1.174	<0.001	1.087	1.025–1.154	0.006	1.100	1.035–1.168	0.002
FFS	2.208	1.543–3.160	<0.001	1.282	0.829–1.981	0.264	1.383	0.880–2.173	0.160
ESR	1.009	1.000–1.018	0.058	0.990	0.976–1.004	0.162	0.989	0.976–1.003	0.135
CRP	1.008	1.003–1.014	0.004	1.007	0.998–1.015	0.131	1.006	0.998–1.014	0.170
BARD score ≥ 2	6.307	0.855–46.514	0.071	6.432	0.796–51.974	0.081			
BARD score ≥ 3	3.267	1.421–7.512	0.005				2.866	1.175–6.991	0.021

AAV: ANCA-associated vasculitis; ANCA: antineutrophil cytoplasmic antibody; MPO: myeloperoxidase; P: perinuclear; PR3: proteinase 3; C: cytoplasmic; BVAS: Birmingham vasculitis activity score; FFS: five-factor score; ESR: erythrocyte sedimentation rate; CRP: C-reactive protein; BARD: BMI, AST/ALT ratio, and T2DM; BMI: body mass index; AST: aspartate aminotransferase; ALT: alanine aminotransferase; T2DM: type 2 diabetes mellitus.

## Data Availability

The data used to support the findings of this study are included within the article and the Appendix A.

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
