# Peer review of "Clinical Link between the BARD Score at Diagnosis and Mortality during Follow-Up in Patients with Antineutrophil Cytoplasmic Antibody-Associated Vasculitis"

_jcm, 2023, doi:10.3390/jcm12175679_

Round 1

Reviewer 1 Report

major:

1. Although the study mentioned many times... age, and sex, it is hard to find any material and methods, results, and/or discussion something about females.

2. line 27: The median age of AAV patients was 60.0 years, and 34.7% were men, WHERE IS THE REST PATIENTS (65.3%, FEMALES)???

3. THE FEMALE WORD IS COMPLETELY MISSED IN THE ENTIRE MANUSCRIPT???!

4. Table 1. Characteristics of AAV patients at diagnosis and during follow-up (N=236) dedicated for 82 male... OK but what about the rest number ??? nothing there. similarly, table 2 too.

minor:

1. Do authors always use male sex? the male is indicative!

2. line 72: VasculitidEs into Vasculitides

3. line 114-115: is this author's definition OR institutional/international definition? PLEASE clarified.

4. lines 137-138: The median BARD score was 2.0, and 189... to be understandable sentence should be The median BARD score was 2.0, WHILE 189...

5. 

Author Response

Reviewer (1)’s comments

Manuscript number: jcm-2580475

Title: Clinical link between the BARD score at diagnosis and mortality during follow-up in patients with antineutrophil cytoplasmic antibody-associated vasculitis

We appreciate your excellent review of our manuscript. Your valuable comments helped us to make a better revision.

Major comments

  1. Although the study mentioned many times... age, and sex, it is hard to find any material and methods, results, and/or discussion something about females.

             Because the male sex has been known as a risk factor for mortality and in this study, it tended to be associated with all-cause mortality in the univariable Cox analysis, we described the male sex.

             We agree with your indication and added the data regarding the female sex to the text in the METHODS, RESULTS sections, Tables 1 and 2 as below:

“Regarding variables at AAV diagnosis, age, male and female sexes, BMI, and ex-smoker were collected as demographic data, and AAV subtype, ANCA type, BVAS, and FFS were collected as AAV-specific data.” (Lines 92-94)

“Regarding variables at AAV diagnosis, the median age of AAV patients was 60.0 years. Of the 236 patients, 34.7%, and 65.3% were males and females.” (Lines 132-133)

Table 1. Characteristics of AAV patients at diagnosis and during follow-up (N=236)

Variables

Values

At AAV diagnosis

Demographic data

  Age (years)

60.0 (48.3-69.0)

  Male sex (N, (%))

82 (34.7)

  Female sex (N, (%))

154 (65.3%)

“In the univariable Cox analysis, both the BARD score ≥2 (HR 6.307, P = 0.071), and the BARD score ≥3 (HR 3.267, P = 0.005) were significantly associated with all-cause mortality during follow-up, along with age (HR 1.086, P <0.001), male and female sexes (HR 2.094, and 0.478, respectively, P = 0.053), BVAS (HR 1.117, P <0.001), FFS (HR 2.208, P <0.001), ESR (HR 1.009, P = 0.058), and CRP (HR 1.008, P = 0.004). Although both male and female sexes exhibited a tendency to be associated with all-cause mortality, because male sex showed a proportional association compared to an inverse association of female sex, male sex was selected and included in the multivariable Cox analysis.” (Lines 166-174)

Table 2. Cox hazards model analyses of variables at diagnosis for all-cause mortality during follow-up in AAV patients

Variables

Univariable

Multivariable

(BARD score ≥2)

Multivariable

(BARD score ≥3)

HR

95% CI

P value

HR

95% CI

P value

HR

95% CI

P value

Age (years)

1.086

1.044-1.130

<0.001

1.060

1.015-1.108

0.009

1.061

1.015-1.109

0.009

Male sex

2.094

0.989-4.431

0.053

2.698

1.181-6.160

0.019

2.327

1.035-5.233

0.041

Female sex

0.478

0.226-1.011

0.053

  1. line 27: The median age of AAV patients was 60.0 years, and 34.7% were men, WHERE IS THE REST PATIENTS (65.3%, FEMALES)???

             According to your comment, we amended the text in the RESULTS section as below:

““Regarding variables at AAV diagnosis, the median age of AAV patients was 60.0 years. Of the 236 patients, 34.7%, and 65.3% were males and females.” (Lines 132-133)

  1. THE FEMALE WORD IS COMPLETELY MISSED IN THE ENTIRE MANUSCRIPT???!

             We added the data regarding the female sex to the text in the METHODS, RESULTS sections, Tables 1 and 2, as in the answer to comment 1.

  1. Table 1. Characteristics of AAV patients at diagnosis and during follow-up (N=236) dedicated for 82 male... OK but what about the rest number ??? nothing there. similarly, table 2 too.

             According to your comment, we added the data regarding female sex in Tables 1 and 2 as below:

Table 1. Characteristics of AAV patients at diagnosis and during follow-up (N=236)

Variables

Values

At AAV diagnosis

Demographic data

  Age (years)

60.0 (48.3-69.0)

  Male sex (N, (%))

82 (34.7)

  Female sex (N, (%))

154 (65.3%)

Table 2. Cox hazards model analyses of variables at diagnosis for all-cause mortality during follow-up in AAV patients

Variables

Univariable

Multivariable

(BARD score ≥2)

Multivariable

(BARD score ≥3)

HR

95% CI

P value

HR

95% CI

P value

HR

95% CI

P value

Age (years)

1.086

1.044-1.130

<0.001

1.060

1.015-1.108

0.009

1.061

1.015-1.109

0.009

Male sex

2.094

0.989-4.431

0.053

2.698

1.181-6.160

0.019

2.327

1.035-5.233

0.041

Female sex

0.478

0.226-1.011

0.053

Minor comments

  1. Do authors always use male sex? the male is indicative!

             We agree with your indication and added the data regarding the female sex to the text in the METHODS, RESULTS sections, Tables 1 and 2 as mentioned above.

  1. line 72: VasculitidEs into Vasculitides

             According to your indication, we amended the phrase as below:

“the Severance Hospital ANCA-associated Vasculitides (SHAVE) cohort” (Lines 71-72)

  1. line 114-115: is this author's definition OR institutional/international definition? PLEASE clarified.

             In the present study, the authors proposed the definition of all-cause mortality and amended the text in the METHODS section as below:

“In the present study, we defined all-cause mortality as death of any cause.” (Line 114)

  1. lines 137-138: The median BARD score was 2.0, and 189... to be understandable sentence should be The median BARD score was 2.0, WHILE 189...

             As you recommended, we amended the text in the RESULTS section as below:

“The median BARD score was 2.0, while 189 (80.1%) and 40 (16.9%) patients had the BARD score ≥ 2 and ≥ 3, respectively.” (Lines 138-139)

Reviewer 2 Report

In this study, Lee et al investigated whether the BARD score at diagnosis could predict all-cause mortality in patients with antineutrophil cytoplasmic antibody-associated vasculitis (AAV). This study included immuno-suppressive-drug naïve 236 patients without chronic liver diseases such as viral hepatitis, non-alcoholic fatty liver disease (NAFLD), and advanced liver diseases, and their clinical data at diagnosis were collected such as age, sex, and Birmingham Vasculitis Activity Score (BVAS). The BARD score was calculated by the sum of aspartate transaminase (AST)/alanine transaminase (ALT) ratio ≥0.8 (+2 points), body mass index (BMI) ≥28 kg/m2 (+1 point), and the presence of type 2 diabetes mellitus (T2DM) (+1 point). All-cause mortality was investigated as the poor outcome of AAV. The median age of AAV patients was 60.0 years, and 34.7% were men. Among AAV patients, 7, 50, and 187 scored 1, 1, and 2 points owing to having BMI ≥ 28kg/m2, T2DM, and AST/ALT ratio ≥ 0.8, respectively. Patients with the BARD score ≥2 and those with the BARD score ≥3 exhibited significantly lower cumulative patients’ survival rates than those without  (P = 0.038 and P = 0.003, respectively). In the multivariable Cox analysis, among the two cut-offs of the BARD scores, only the BARD score ≥3 (HR 2.866) along with age (HR 1.061), male sex (HR 2.327), and BVAS (HR 1.100), was independently associated with all-cause mortality during follow-up. Authors concluded that the BARD score ≥3 at AAV diagnosis could predict all-cause mortality during follow-up in AAV patients. The conclusion that the BARD score, a score originally used for NAFLD, is an independent prognostic factor for AAV is very interesting. One limitation of this study is that no echocardiography or other tests were performed during screening. In practice, it is difficult to perform all the tests, so the conclusion that the BARD score, which is a simple score, is useful in predicting prognosis may be useful in real practice.

I have a question as follows.

major concerns)

1) In this article, only all cause mortality is examined. However, since the patients were being treated with immunosuppressive drugs, various causes of death can be considered, including bacterial or viral infections, as well as worsening of the disease. Therefore, it would be very easy for readers to understand if all causes of death are listed in a table by type. Also, depending on the cause of death, the message conveyed to the reader may differ. For example, if the number of infections is high and the BARD score is high, the message would be that attention to infections is necessary because the risk of suffering from infections is considered to be high even if the same treatment is given. Conversely, if the patient often dies from worsening disease, a high BARD score would be a consideration that the disease is considered to be more aggressive and therefore stronger treatment is needed.

Author Response

Reviewer (2)’s comments

Manuscript number: jcm-2580475

Title: Clinical link between the BARD score at diagnosis and mortality during follow-up in patients with antineutrophil cytoplasmic antibody-associated vasculitis

We appreciate your excellent review of our manuscript. Your valuable comments helped us to make a better revision.

Major concerns

In this article, only all-cause mortality is examined. However, since the patients were being treated with immunosuppressive drugs, various causes of death can be considered, including bacterial or viral infections, as well as worsening of the disease. Therefore, it would be very easy for readers to understand if all causes of death are listed in a table by type. Also, depending on the cause of death, the message conveyed to the reader may differ. For example, if the number of infections is high and the BARD score is high, the message would be that attention to infections is necessary because the risk of suffering from infections is considered to be high even if the same treatment is given. Conversely, if the patient often dies from worsening disease, a high BARD score would be a consideration that the disease is considered to be more aggressive and therefore stronger treatment is needed.

             We totally agree with your recommendation which could make the clinical significance of the results of the present study stronger. However, in real clinical practice, it is often difficult to clearly distinguish the cause of death in deceased patients with AAV: for instance, in a case of death from septic shock caused by infection while increasing the dose of immunosuppression due to worsening of the disease, which of the following is the most appropriate cause of death among the worsening of the disease, side effects of immunosuppressive treatment, and septic shock caused by infection? Owing to these difficult clinical situations, in the present study, we defined all-cause mortality as death of any cause.

             If you do not mind, I would like to add this context to the LIMITATION section as below:

“Additionally, further investigation of the association of the BARD score with death according to the causes of death may clarify the clinical usefulness of the BARD score and enhance its significance in real clinical settings. However, we could not perform it due to the clinical limitation that it is often difficult to clearly distinguish the causes of death in deceased patients with AAV among the worsening of the disease, side effects of immunosuppressive treatment, and serious infections. This was why we defined all-cause mortality as death of any cause in the present study.” (Lines 290-297)

Round 2

Reviewer 1 Report

Thank you very much for considering our comments!

Reviewer 2 Report

No additional comments. Authors responded my questions adequately.